# Correlation between Microstructure and Hydrogen Degradation of 690 MPa Grade Marine Engineering Steel

**DOI:** 10.3390/ma14040851

**Published:** 2021-02-10

**Authors:** Heng Ma, Huiyun Tian, Juncheng Xin, Zhongyu Cui

**Affiliations:** 1Yinshan Section Steel Corporation of Laiwu Steel Group Ltd., Jinan 271104, China; maheng418@126.com; 2School of Materials Science and Engineering, Inorganic Materials Faculty, Ocean University of China, Qingdao 266100, China; junchengouc@163.com (J.X.); cuizhongyu@ouc.edu.cn (Z.C.)

**Keywords:** low-alloy steel, heat treatment, cyclic voltammetry, hydrogen permeation, HIC

## Abstract

Electrochemical H charging, hydrogen permeation, and hydrogen-induced cracking (HIC) behavior of 690 MPa grade steel substrate and different heat-treatment states (annealed, quenched, normalized, tempered) are investigated by cyclic voltammetry (CV), hydrogen permeation, electrochemical H charging, and slow strain rate tensile test (SSRT). The results show that hydrogen diffuses through the steel with the highest rate in base metal and the lowest rate in annealed steel. The hydrogen-induced cracks in base metal show obvious step shape with tiny cracks near the main crack. The cracks of annealed steel are mainly distributed along pearlite. The crack propagation of quenched steel is mainly transgranular, while the hydrogen-induced crack propagation of tempered steel is along the prior austenite grain boundary. HIC sensitivity of base metal is the lowest due to its fine homogeneous grain structure, small hydrogen diffusion coefficient, and small hydrogen diffusion rate. There are many hydrogen traps in annealed steel, such as the two-phase interface which provides accommodation sites for H atoms and increases the HIC susceptibility.

## 1. Introduction

Hydrogen in microstructures can be roughly divided into two forms, diffusing hydrogen and trapping hydrogen [1]. The diffusion rate and solubility of hydrogen in steel substrates are influenced by hydrogen traps, which can enhance or decrease the hydrogen-induced-cracking (HIC) sensitivity of the steel. Pressouyre [2] classified hydrogen traps as reversible and irreversible traps according to the desorption activation energy of hydrogen, which can be measured by thermal desorption spectroscopy. If the *E*_a_ (binding energy) of the trap is higher than 50 kJ/moL, the trap is irreversible and can capture hydrogen until it reaches the saturated state [3]. When the temperature rises or the hydrogen content exceeds the saturation concentration, the hydrogen leaves and diffuses into the lattice. The *E*_a_ of reversible traps, which can easily capture and release hydrogen even at low temperatures, is lower than 30 kJ/mol [4]. One important factor affecting HIC sensitivity is the amount of hydrogen in steel [5]. The HIC sensitivity of steel may increase with increasing hydrogen concentration both internally and externally. However, the hydrogen content that causes HIC may be affected by other factors. For example, the critical value of hydrogen content that causes HIC may be affected by the applied stress, microstructure, and tensile strength [6,7,8]. Moreover, when the hydrogen content in steel reaches a saturation concentration, the HIC sensitivity of steel will not change significantly. Hydrogen capture has a great influence on hydrogen accumulation and mobility in steel, and the microstructure of steel can capture and limit hydrogen.

The HIC sensitivity of steels with different microstructures is also different, because of the difference in the distribution of phases, grain size, and defects, which can affect hydrogen diffusion and accommodation [9]. When tensile strength is lower than 850 MPa, tempered steel is almost unaffected by HIC unless hydrogen content is approximately 10 ppm [10]. However, martensitic high-strength steels with a tensile strength of 900–1400 MPa are highly susceptible to HIC at hydrogen concentrations as low as 0.5–1 ppm. Quenched martensite is most prone to hydrogen embrittlement [11], which may be due to martensite that produces high residual stress in the lattice. Quadrini et al. [12] also concluded that the thermodynamic stable phase was not easily affected by hydrogen embrittlement. This seems to explain why tempering treatment can improve the resistance of martensitic steel to hydrogen embrittlement [13]. However, if only martensite exists in the steel microstructure, this does not necessarily mean high hydrogen embrittlement sensitivity. Many other factors, such as inclusion, alloy content, second phase precipitation, grain structure, and martensite type, can affect the hydrogen embrittlement sensitivity in martensitic steel [11]. The rate of hydrogen diffusion in austenite steel, which has a dense lattice structure, is about three to four orders of magnitude lower than that in martensite steel, which means that it takes a longer time for hydrogen to accumulate in the austenite and reach its critical value. Mohammadi et al. [14] found that the lower the heat treatment temperature, the more M-A island components. There may be many potential hydrogen capture sites in the heat-treated steel, and HIC occurs easily [15,16]. For dual-phase steel consisting of ferrite and martensite phases, hydrogen may be trapped at the phase interface, grain boundaries, dislocations, and lattice vacancies [17,18]. The microstructure of some low-carbon steels is mainly composed of a ferrite and fine pearlite phase in which hydrogen can be enriched [19,20]. At present, a lot of studies have been carried out on the HIC behavior of microstructures, but researches on how to quantitatively analyze the relationship between microstructure and hydrogen are in progress.

So far, the quantitative information of diffused hydrogen in steel can be obtained using electrochemical H charging, hydrogen permeation, cyclic voltammetry (CV), thermal extraction, and other testing methods [1,21,22,23,24,25]. By comparing the microstructure characteristics of the steel substrate and heat-treated steel after different heat treatments, the interaction mechanism between hydrogen and microstructure is evaluated. In this paper, the relationship between the adsorption/desorption behavior of diffused hydrogen and the H-charging time was studied by electrochemical methods based on CV and potentiostatic polarization. The quantitative analysis of diffused hydrogen was carried out by electrochemical H-discharge and hydrogen permeation. The influence of microstructure on HIC sensitivity was verified by the HIC test and slow strain rate tensile test.

## 2. Experimental

### 2.1. Materials

The materials used in this work were cut from a thick plate of 690 MPa grade steel with a chemical composition (wt.%) of 0.14 C, 0.035 Si, 1.31 Mn, 0.51 Cr, 0.02 Ni, 0.134 Mo, 0.024 Nb, 0.004 V, and 0.001 B and balanced by Fe. The nominal yield strength and tensile strength of this steel are about 690 MPa and 720 MPa, respectively. After heating the steel to 1150 °C for 10 min in a muffle furnace, the steel was annealed, quenched, normalized, and tempered respectively to obtain the steels with different microstructures. The specimens with a dimension of 10 mm × 10 mm × 2.0 mm were used to heat treatment. High-temperature tempering was used in tempering heat treatment, and the tempered temperature was 514 °C for 40 min. The heat treatment process of this steel is shown in Figure 1. The atmosphere environment outside the heat treatment furnace was about 22 °C at room temperature, and the relative humidity was about 50%. The atmospheric environment in the heat treatment furnace was air.

### 2.2. Microstructure Characterization and Microhardness Test

The steel substrate and heat-treated specimens were ground to 5000 # waterproof sandpaper successively, polished with a 1.5 μm diamond grinding paste, cleaned with ethanol and acetone, rinsed with deionized water, and dried in air for microstructure observation. The steel substrate and heat-treated specimens were etched with 4 vol.% nital solution for 10 s and saturated picric acid containing a small amount of detergent at 70 °C for 3–5 min to observe the microstructure and prior austenite grain boundary, respectively, with a light microscope (Zeiss Lab. A1 (Zeiss, Oberkochen, Germany)) and scanning electron microscope (SEM, Quanta 250, Portland, OR, USA).

The Vickers microhardness profile of the steels was measured by automatic turret digital display microhardness tester with image analysis (JMHVS-1000ZCCD Shanghai Precision Instrument Co., Ltd, Shanghai, China). The different microstructure sites were observed through a microscope and were subjected to continuous hardness tests, with a load of 100 g for 10 s. At least 40 sites were detected to get reproducible values.

### 2.3. Electrochemical Test

The specimens (10 mm × 10 mm × 2 mm) used for the electrochemical tests were encapsulated with epoxy, leaving the surface of 1 cm^2^ as the working electrode [26]. The specimens were ground sequentially to 1500 # by an emery paper and then degreased in dehydrated ethanol, ultrasonic rinsed with deionized water, and dried in air.

The electrochemical tests were conducted through an electrochemical workstation (Autolab PGSTAT 302 N Metrohm Autolab B.V., Utrecht, The Netherlands) with the three-electrode system, in which the platinum was the counter electrode and a saturated Ag/AgCl electrode (+0.197 V vs. SHE) was the reference electrode. The electrolyte solution is 1 mol/L NaOH solution containing 8 g/L of thiourea. Many researchers have found that thiourea can act as a hydrogen recombination inhibitor to prevent H^+^ binding and escaping from the specimen surface [27,28]. The NaOH solution was chosen to prevent corrosion of the steels during electrochemical experiments.

The cyclic voltammetry adopted in this paper consisted of three steps:

Step 1: Surface pretreatment of specimens before electrochemical H charging was conducted, namely, two consecutive CV scans, recording the current and potential curves, to analyze and judge the reaction mechanism of the specimen surface.

Step 2: The potential of −1.25 V (vs. Ag/AgCl) was applied to the specimen for cathodic polarization, and H charging was carried out by electrolytic water reaction (Equation (1)) on the working electrode.
(1)$$2H_{2}O\overset{\mathrm{electrolysis}}{\to }2H_{2}↑+O_{2}↑$$

The current transient curves were recorded under potentiostatic polarization H charging to assess the current response and determine the charge as a function of time. The H charging was performed with different times (10 min, 30 min, 1 h, 2 h, 3 h, and 4 h) to compare the hydrogen saturation levels of the different microstructures.

Step 3: Three CV scans were recorded after the H-charging step. For most experiments, the potential started at −1.25 V (vs. Ag/AgCl) with a scanning rate of 10 mV/s.

The cyclic voltammetry method proposed and validated by Ozdirik et al. [1] includes the H-discharging step in addition to the H-charging step. The CV method can be used to monitor the relationship between the adsorption/desorption behavior of diffused hydrogen and the hydrogen charging time, and to determine the hydrogen saturation level of the different microstructures. Additionally, special H-discharging experiments are required for the quantitative analysis of diffused hydrogen in an electrochemical test. The H-discharging experiment has one more step than the H-charging experiment, that is, the potentiostatic polarization H-discharging (at −0.9 V vs. Ag/AgCl for 30 min) is carried out before the H-charging step 3. The purpose of this experimental procedure was to verify the adsorption/desorption of hydrogen and to compare the amount of accumulated and released hydrogen during charging and discharging between pure NaOH and NaOH solution containing thiourea. By integrating the current recorded during the H-discharging experiment with the current transient curves (only the first 100 s of discharge), the quantity of electric charge after H-discharging (*Q*_H.charg_) and the background quantity of electric charge (*Q*_Non.H_) could be obtained. After two CV scans (step 1), a constant potential of −0.9 V (vs. Ag/AgCl) was immediately applied for H-discharging, and the transient current recorded during the period was the background current. Therefore, the background quantity of electric charge (*Q*_Non.H_) of the steel was about 3.5 × 10^−3^ C/cm^2^ through the current-time transient integral.

According to Faraday’s formula (Equation (2)), the concentration of absorbed hydrogen (*C*_0_) that is oxidized during discharging can be calculated as follows:(2)$$C_{0}=\frac{Q_{H.\;charg}-Q_{Non.\;H}}{nFV}$$
where *n* is the number of electrons involved in the oxidation reaction (H→H^+^ + e), *F* is Faraday’s constant (96,485 C/mol), *V* is the effective volume of the specimen (cm^3^). The effective volume of all specimens used in the electrochemical H-charging experiment is 0.02 cm^3^.

The effect of thiourea on the electrochemical behavior of the steel at cathodic potential was studied by the linear scanning volt-ampoule method (LSV). The experiment began with two consecutive CV scans as described above (step 1), and then a linear scan was performed in NaOH solution with and without thiourea at a scanning rate of 2 mV/s from open circuit potential (OCP) to −1.6 V (vs. Ag/AgCl).

### 2.4. Hydrogen Permeation Test

According to ASTM G148, the Devanathan–Stachurski cell, which consists of two compartments including the charging cell and the oxidation cell, was used in the hydrogen permeation test. The hydrogen permeation specimens were rectangular thin slices with a thickness of 1.5 mm, leaving a rounded exposed area of 1.5 cm^2^ as the working surface. To ensure stable oxidation, the specimens were polished with a 1.5 μm diamond grinding paste. Afterward, constant current nickel plating with a galvanostatic current density of 3 mA/cm^2^ for 10 min in a Watt coating bath (250 g/L NiSO_4_·6H_2_O + 45 g/L NiCl_2_·6H_2_O + 40 g/L H_3_BO_3_) was conducted. The charging side of the specimen was only ground up to 1500 # and then cleaned with deionized water. To avoid the influence of hydrogen produced during specimen preparation on the testing results, the specimen was placed in a vacuum drying oven at 150 °C for 24 h.

To fully oxidize the diffused hydrogen atoms on the oxidizing side, the specimens were imposed a potential of +300 mV (vs. SCE saturated calomel electrode) in deaerated NaOH (0.2 mol/L). After the background current density dropped below 0.1 μA/cm^2^, H-charging solution (1 mol/L NaOH solution containing 8 g/L of thiourea) was introduced into the charging cell and H-charging potential of −1.25 V (vs. Ag/AgCl) was applied to the specimen. Hydrogen permeation tests of five microstructures were repeated at least three times at room temperature (22 °C).

The hydrogen permeation parameters were calculated from the obtained permeation curves by Equations (3)–(5).
(3)$$D_{eff}=\frac{L^{2}}{6t_{lag}}$$
(4)$$J=\frac{i_{ss}}{nF}$$
(5)$$C_{app}=\frac{J\times L}{D_{eff}}$$
where *D_eff_* (cm^2^/s) is the effective diffusion coefficient, *J* is hydrogen flux (mol/(cm^2^·s)), *C_app_* is apparent hydrogen concentration (mol/cm); *i_ss_* is the steady-state permeation current, *L* is the specimen thickness, *t_lag_* means the time when the steady-state current value is multiplied by 0.63.

### 2.5. Hydrogen-Induced Cracking Test

The location and propagation path of hydrogen-induced cracks in different microstructures were observed. For this test, the six surfaces of the specimen (75 mm × 40 mm × 12 mm) were ground to 5000 # with waterproof sandpaper, cleaned with acetone, and dried in air. Then, the samples were electrochemically charged in deaerated 0.5 mol/L H_2_SO_4_ for 12 h at −1.25 V (vs. Ag/AgCl). After H charging, the specimen was ground to 5000 #, polished with diamond polishing paste, and etched with 4 vol.% nital solution. Then, the crack initiation and propagation were observed with a metallographic microscope.

### 2.6. Slow Strain Rate Tensile Test (SSRT)

The SSRT method was used to investigate the hydrogen-induced cracking behavior of the steel with different microstructures in H-charging solution (1 mol/L NaOH solution containing 8 g/L of thiourea) at −1.25 V (vs. Ag/AgCl). According to GB/T 15970, SSRT specimens were prepared and then ground with sandpaper along the stretch direction to 1500 # on the working surface. Before the test, the specimens were pre-charged for 12 h to ensure steady-state surface conditions and a uniform hydrogen concentration. A tensile rate of 0.0018 mm/min (yielding a strain rate of 10^−6^/s) was used to carry out the SSRT tests with the WDML-30KN Material Test System (manufacturer, city, country). The SSRT results under each condition were tested three times to check the reproducibility.

To quantitatively characterize the cracking susceptibilities of the steel with different microstructures, the sensitivity in terms of elongation loss rate (*I_δ_*) was calculated using the following equation [29]:(6)$$I_{\delta }=\left(1-\frac{\delta_{s}}{\delta_{0}}\right)\times 100\%$$
where *δ_s_* and *δ*_0_ are the elongation of steel in test solutions and air, respectively.

## 3. Results and Discussion

### 3.1. Microstructure

Figure 2 shows the microstructure characteristics of the 690 MPa grade steel with different heat-treatment states. The austenitizing temperature was chosen at 1150 °C for 10 min because the microstructure difference was obvious and relatively clear at this temperature. Ma et al. [30] found that the critical transition temperature of E690 steel in re-austenitizing along the prior-austenitic grain boundary is about 745 °C. Farzad et al. [14] reported that the lower the heat treatment temperature, the more M-A island components existed in API X80 pipeline steel. According to the previous experimental results, the critical quenching temperature for the roughening of quenched martensite microstructure of the steel is around 1150 °C (not shown).

The microstructure of the steel substrate consists of granular bainite (GB) and a small amount of lath bainite (LB), with fine and uniform grains (Figure 2a,b). The microstructure after annealing treatment consists of ferrite and pearlite, which exhibit white and black color in the light microscope image (Figure 2c) and vice versa in the SEM picture (Figure 2d). Five different fields of view were selected to analyze the ratio of two phases after annealing through binary extraction in metallographic analysis software. The results show that the volume fractions of ferrite and pearlite are 71% and 29%, respectively. The microstructure of the steel after quenching treatment consists of lath martensite (LM) and a small amount of bainite (Figure 2e,f). The white network at the grain boundary is ferrite and there are featherlike bainite (FB) structures. Martensite packet is defined as a lath structure with the same surface, which is composed of lath or packet [31]. With the further study of martensite substructure by researchers, martensite lath block is generally considered as a lath structure with a similar orientation. The microstructure of the steel after normalizing consists of granular low-carbon bainite (GLCB), which is an island structure composed of ferrite and cementite (Figure 2g,h). After tempering heat treatment, the microstructure consists of tempered sorbite. The gray area is fine acicular martensite, and the black part is sorbite in the SEM image (Figure 2n).

Figure 3 shows the morphology of prior austenite grains of the steel with different heat treatment states. The effective grain size is an important parameter to indicate the anti-crack ability of high-strength low-alloy steel [31]. The average size of prior austenite grains of the original microstructure and the heat-treated steels was quantitatively analyzed by using the straight-line transversal method. For the accuracy of the statistical results, the number of prior austenite grains in each heat-treated steel should be counted at least 200. The results are shown in Figure 4.

The metallographic and statistical results of the prior austenite grains show that the steel substrate has the finest grain with an average grain diameter of 16.7 μm. Annealed steel, quenched steel, normalized steel, and tempered steel have a similar grain size. Due to the different positions of samples in the muffle furnace during heat treatment, the grain size has some fluctuations in the acceptable range. Because annealed steel has no prior austenite grain boundary, ferrite grain size was calculated.

Microhardness is an important parameter to evaluate the resistance of metal to local deformation. It can be seen from Figure 4b that the microhardness of quenched steel is the highest (402 HV), which conforms to the characteristics of quenched martensite. The microhardness of normalized steel is relatively lower because of the existence of a large amount of ferrite (195 HV). The average Vickers microhardness of base metal, normalized steel, and tempered steel is 282 HV, 295 HV, and 326 HV, respectively.

### 3.2. Electrochemical Behavior

#### 3.2.1. Effects of Thiourea on the Electrochemical Behavior

Figure 5 shows the linear sweep voltammogram of base metal for NaOH solution with and without thiourea and a larger view in LSV at −1.25 V (vs. Ag/AgCl). As shown in Figure 5, a higher overpotential is required to produce the same current in the thiourea-containing NaOH solution than that in the pure NaOH solution. At −1.25 V (vs. Ag/AgCl), the current in the pure NaOH solution is approximately four times higher than that in the thiourea-containing NaOH solution. In addition, in the pure NaOH solution, the formation of bubbles can be observed on the specimen surface when the potential is more negative than −1.25 V (vs. Ag/AgCl) and bubbles completely cover the specimen surface as the potential is shifted lower than −1.4 V (vs. Ag/AgCl). However, in the solution containing thiourea, no bubble formation was observed above −1.4 V (vs. Ag/AgCl). From −1.4 V (vs. Ag/AgCl) to −1.6 V (vs. Ag/AgCl), slight bubble formation was detected which is much less than that in pure NaOH. This further proves that the hydrogen release reaction is inhibited by thiourea. Ozdirik et al. [1] proved that thiourea inhibits the recombination of H atoms to form H_2_ when they studied hydrogen adsorption/desorption of SAE 1010 steel. Some researchers have shown that thiourea prevents the recombination of H atoms to form hydrogen, but promotes the entry of H atoms into the steel [27,32,33].

Figure 6 shows the cyclic voltammograms (CVs) of the base metal before and after H charging in NaOH solution with and without thiourea.

The results show that there are two anodic reactions before and after hydrogen charging in the thiourea-containing NaOH solution, as shown in Figure 6a. The first anodic reaction peak before H charging is labeled as peak a (at −0.87 V (vs. Ag/AgCl)) and the first anodic reaction peaks after H charging are labeled as peak a.1 (at −0.92 V (vs. Ag/AgCl)) and a.2 (at −0.87 V (vs. Ag/AgCl)). In NaOH solution without thiourea, the anodic reaction peak (namely peak a′ at −0.92 V (vs. Ag/AgCl)) also changes significantly after hydrogen charging (Figure 6b). It is worth noting that the current density in the thiourea-containing NaOH solution is much higher than that in pure NaOH. As shown in Figure 6, in thiourea-containing NaOH solution, peak a.1 only appears after H charging, and peak a.1 appears at the same potential as peak a’ in pure NaOH solution. This indicates that the position of this peak is related to H charging in both solutions. In addition, in the thiourea-containing NaOH solution, the potential of peak a.2 is consistent with that of peak a before H charging, while in the pure NaOH solution, no peak appears at this potential, which proved that peak a.2 is a reaction related to thiourea.

#### 3.2.2. Effects of H charging on the Electrochemical Behavior

Figure 7 shows the cyclic voltammograms of the 690 MPa grade steel substrate before (a) and after H charging for 30 min (b) in NaOH solution containing thiourea.

Two consecutive CV scans were performed on the base metal as a pretreatment before H charging, as shown in Figure 7a. Two anodic and one cathodic response can be observed in the CV curves (Figure 7a). The current density value of peak a decreases with the number of scans, while those of peak b and peak c increase with the cycle. The results of the area integrals of peak b and peak c show that the quantity of electric charge released by the two peaks is similar. The color of the specimen changed to brownish-yellow during the anodic scan (starting from −0.8 V (vs. Ag/AgCl)) and disappeared again during the reverse scan. Peaks b and c in the cyclic voltammetry are usually attributed to hydrogen oxidation/reduction reactions [34,35,36]. In Figure 7b, three continuous CV curves recorded after electrochemical H charging are shown. The first anodic response in the first CV scan was marked as two peaks of a.1 and a.2. In the second CV scan, peak a.1 disappeared while peak a.2 continued to exist. However, it can be seen from Figure 7b that there was a gradual downward trend of peak a.2 current density from the first scan to the third scan. From the third scan, there was no significant change in peak a.2 [1]. The potential of the peak a before H charging was the same as that of peak a.2 after H charging. The current density values of peak b and peak c increased with the number of scans, which may be related to the formation of multilayer iron oxide or the surface roughness of the specimen [37].

To clarify the relationship between H-charging time and peak of CV, the first CV scanning curve of the base metal after H charging is shown in Figure 8.

The shape and height of peak a.1 and peak a.2 depend on the duration of H charging. After 10 min of H charging, peaks a.1 and a.2 are nearly the same height in the first CV scan. After H charging for 30 min, the current density values of peak a.1 and peak a.2 increase with a more obvious increasing trend of the former peak. As the H-charging time continues from 1 to 3 h, the heights of peaks a.1 and a.2 show little change. The results show that the peaks a.1 and a.2 are related to the H-charging time, and the steel substrate almost reaches the hydrogen saturation state after the H charging for 1 h. In addition, under all H-charging time conditions, the three consecutive CV curves after H charging show that the current density value of peak a.2 tends to decrease from one scan to the next. Moreover, peak a.1 did not appear again during the second CV scan, indicating that hydrogen was completely desorbed from the steel during the first CV scan, independent of the scanning rate and the H-charging time [1,27]. The potential corresponding to peak a.1 was −0.92 V (vs. Ag/AgCl), which was taken as the discharge potential in the potentiostatic polarization H-discharging experiment. In three consecutive CV scans, the peak a.2 decreased but did not disappear. Since peak a.2 only existed in the CV curve of thiourea-containing NaOH solution, it is speculated that it must be an oxidation process related to thiourea [1].

#### 3.2.3. Effects of Microstructure on the Electrochemical Behavior

Figure 9 shows the first CV scans for five different microstructure specimens after hydrogen charging for 30 min in NaOH solution containing thiourea.

The CV curves mainly included peak a and peak b, wherein peak a was further divided into peak a.1 and peak a.2 which are attributed to hydrogen oxidation and thiourea-related reactions, respectively. The current density value of peak b in CV curves was very similar in the five microstructures, while significant differences could be seen in peak a. Though, after H charging after 30 min, five different microstructures were similar in oxidation peak shape, but the current density value of peak a was the highest in the tempered steel and the lowest in the normalized steel. This may be related to hydrogen permeation in the microstructure [38].

As discussed above, peak a can be viewed as a function of H-charging time. The base metal reached the hydrogen saturation state after H charging for 1 h as indicated by the relatively stable current density of peak a.1 when the charging time exceeded 1 h (Figure 8). Figure 10 shows the first CV scans of the steel with different microstructures after hydrogen charging for different times in NaOH solution containing thiourea.

In the cases of annealed steel, quenched steel, normalized steel, and tempered steel, the current density values of peak a did not change significantly after the H-charging time reached 4 h, 3 h, 3 h, and 2 h, respectively, indicating the accomplishment of the saturation state. The H-charging time required for the annealed steel to reach the hydrogen saturation state was the longest (4 h) and that of the base metal was the shortest (1 h), implying the fast hydrogen diffusion rate in the annealed steel with ferrite and pearlite microstructures. Zhang et al. [39] found that the hydrogen diffusivity within fine granular bainite was higher than that of ferrite for welded X80 steel under pressurized gaseous hydrogen. To further study the relationship between different microstructures and hydrogen behavior, the concentration of absorbed hydrogen (*C*_0_) is calculated by Equation (2) and shown in Figure 11.

The base metal could reach a hydrogen saturation state in a shorter time (1 h), which may be attributed to the number of dislocations and various defects within the fine granular bainite being lower, and thus the diffused hydrogen was not easily trapped and stored. However, the annealed steel has obvious interface and defects and also has a thick pearlite structure, which is convenient for hydrogen capture and storage by diffusion.

### 3.3. Hydrogen Permeation Behavior

The microstructure is an important factor affecting hydrogen permeation and HIC sensitivity [7,38,39]. To accurately indicate the hydrogen absorption and diffusion in five microstructures, the results of hydrogen permeation were analyzed. Figure 12 shows the hydrogen permeation curves and the calculated parameters of different microstructures.

At the charging side, a negative potential of −1.25 V (vs. Ag/AgCl) was applied to reduce hydrogen ions to hydrogen atoms that could be captured by the steel. The thiourea prevented the recombination of H atoms to form H_2_ and promoted the adsorption of H atoms. Then, the hydrogen atoms diffused to the oxidation side across the steel sheet. The diffusion hydrogen oxidized in NaOH solution in the oxidation cell to generate hydrogen ions, and the reaction current was expressed as hydrogen permeation current [40,41]. The hydrogen permeation current density (0.414 × 10^−^^6^ A/cm^2^), hydrogen flux (0.429 × 10^−^^11^ mol/(cm^2^·s)), and apparent hydrogen concentration (0.405 × 10^−^^6^ mol/cm^3^) of the base metal were the lowest, and the effective diffusion coefficient (1.58 × 10^−6^ cm^2^/s) was the highest. Hydrogen atoms had the highest diffusion rate inside the base metal, and those captured by the hydrogen trap were the lowest. Thus, it was easier to reach the saturation state. It was also mentioned in the CV tests that the base metal reached the hydrogen saturation state after H charging for only 1 h. The annealed steel consists mainly of ferrite phases and a small portion of flake pearlite which can be considered reversible hydrogen traps [19,42]. Many researchers believe that hydrogen was preferentially enriched in dislocations, grain boundaries, inclusions, and two-phase interfaces after entering the material [43,44]. Annealed steel has more active sites of hydrogen capture and storage than other microstructures. Hydrogen atoms are more easily pinned into the annealed steel (ferrite and pearlite) [30,38]. In the case of martensitic steel, the martensitic lath boundary and the prior austenite grain boundary are considered as the microstructure characteristics of the hydrogen trap [45,46]. Depover et al. [47] reported that martensitic steel has high-density dislocation in its microstructure, and the content of reversible hydrogen in martensitic steel (at the position error) accounted for about 75% of the total diffused hydrogen content. Therefore, diffused hydrogen in martensitic steel can be derived from martensitic lath boundary, prior-austenite grain boundary, and dislocation. Nava et al. [48] proved that dislocation controlled the effective diffusion coefficient of hydrogen in martensitic steel and the absorption of hydrogen in martensitic steel was mainly due to the contribution of dislocation. Jiang et al. [49] demonstrated that the dislocation walls and cells hindered the diffusion of hydrogen and homogeneous distribution of dislocations dispersed trap sites for capturing hydrogen. Under the action of no external force, the H content of nailing in martensite steel is low due to fewer internal dislocations. Hydrogen permeation results show that annealed steel nailing H is more than quenched steel, but diffusion H is less than quenched steel. The *J* and *C_app_* of annealed steel are slightly smaller than that of quenched steel (Figure 12).

### 3.4. HIC Analysis

Figure 13 shows the light microscope micrographs of the hydrogen-induced cracks formed on the steel with different microstructures after H charging for 12 h in 0.5 mol/L H_2_SO_4_.

In general, hydrogen-induced crack initiation is easy in the “hydrogen trap” of steel, such as inclusions, dislocations, voids, and grain boundaries [50,51,52]. Under the condition of electrochemical H charging, bubbles and surface crack propagation are easy to occur on the surface of the specimen. During the electrochemical H charging, hydrogen atoms are formed on the surface of the specimen, and some hydrogen atoms enter into the steel. Hydrogen molecules are formed by bonding at the hydrogen trap (grain boundary, dislocation entanglement, the second phase, inclusion, etc.) and the high pressure generated in the bonding place will lead to nucleation and propagation of the crack. It can be seen from Figure 13a that the hydrogen-induced crack in the base metal shows an obvious step shape, and tiny cracks are generated near the main crack. If there is an external force, these stepped cracks will perforate under the action of shear stress at both ends, resulting in the reduction of the bearing capacity of the steel structure. Huang et al. [53] reported that low carbon bainite has lower hydrogen cracking sensitivity than quenched and tempered martensite. The crack distribution of annealed steel was mainly along the pearlite (Figure 13b), mainly because the carbide and some inclusions are easy to be precipitated at the pearlite boundary. Hydrogen atoms are easily enriched in these sites, causing hydrogen embrittlement and forming cracks, which expanded under the stress. Some researchers have found that HIC is found to initiate at interfaces of ferrite and pearlite bands [54,55]. Quenched and tempered steel is widely used because of its excellent toughness with high hardness, strength, and weight [9]. These types of steel are prone to HIC, resulting in poor impact resistance [9,56]. The crack of martensite steel mainly propagated by the transgranular method (quenched steel, Figure 13c) and some tiny cracks germinated on the main crack. For the tempered steel, intergranular cracking (IGC) along the prior-austenite grain boundary (PAGB) dominated the fracture process (Figure 13d). IGC along the PAGB could be detected if the H charging was powerful enough, which has been observed in many materials [41,57,58,59].

Figure 14 shows the stress–strain curves of the steel with different microstructures in 1 mol/L NaOH solution containing 8 g/L of thiourea at −1.25 V (vs. Ag/AgCl) (a) and the HIC susceptibility in terms of the elongation loss rate (b).

Due to the influence of hydrogen, the elongation of the base metal and the heat-treated steel in air was significantly higher than that in the NaOH solution. Alvaro et al. [60] reported that the hydrogen in lattice interstitials was mainly responsible for the embrittlement by 3D cohesive modeling. Researchers have reported that the intercritical heat affected zone (ferrite and M-A island microstructure) has a high deformation capacity due to its high ferrite phase proportion and soft phase [30,38]. Although the elongation of annealed steel (ferrite and pearlite microstructure) is the highest, the HIC sensitivity calculated based on the loss of elongation was the largest, which proves that annealed steel is more susceptible to hydrogen. This may have more to do with “hydrogen traps” in annealed and normalized steel [27,31]. Laurent et al. [61] found that residual austenite, pearlite, and bainite grain boundaries could be the active site of pinned hydrogen. The HIC sensitivity of heat-treated steel is higher than that of the base metal (BM) probably due to the existence of a large number of dislocations, phase interfaces and grain boundaries in the heat-treated steel. It was found by Zhang et al. [39] that fine-grain bainite had lower hydrogen embrittlement sensitivity than coarse-grain bainite. Microstructure characteristics, test data, and cracking characteristics for BM and the heat treatment microstructure were summarized and shown in Table 1.

The microstructure morphologies of the base metal and normalized steel were similar, but the grain size of normalized steel was three times higher than that of the base metal. Secondly, apparent hydrogen concentration and hydrogen permeation current density of BM was the lowest. So base metal had the lowest HIC sensitivity in five microstructures. Under the test conditions in this paper, the hydrogen concentration in the steel easily reached the critical value that induces hydrogen embrittlement. When the hydrogen content was high, hydrogen concentration and hydrogen pressure at the grain boundary were relatively high, resulting in the instantaneous cracking along the grain boundary and hydrogen-induced bubbles. HIC sensitivity increased in the following order: fine granular bainite, acicular martensite and tempered sorbite, lath martensite, coarse granular bainite, ferrite, and pearlite.

## 4. Conclusions

The correlation between the microstructure and hydrogen degradation of 690 MPa grade marine engineering steel was investigated in the present work. The following conclusions are obtained:(1)The CV tests show that thiourea is an effective hydrogen permeation accelerator. The hydrogen diffusion rate in the steel base metal with uniform microstructure and fine grain is the highest, while that in the annealed steel with ferrite and pearlite microstructure is the lowest.(2)The hydrogen-induced cracks in the steel base metal show obvious step shape and tiny cracks near the main crack. The cracks of the annealed steel are mainly distributed along pearlite. The crack propagation of martensite steel (quenched steel) is mainly transgranular, while the cracks of tempered steel along the prior austenite grain boundary.(3)HIC sensitivity of the base metal is the lowest due to its low hydrogen flux and apparent hydrogen concentration. Annealed steel exhibits higher HIC sensitivity at lower hydrogen diffusion flux and surface hydrogen concentration, due to many hydrogen traps in annealed steel. Annealed steel with a ferrite and pearlite microstructure is more susceptible to hydrogen.

## Figures and Tables

**Figure 1 materials-14-00851-f001:**
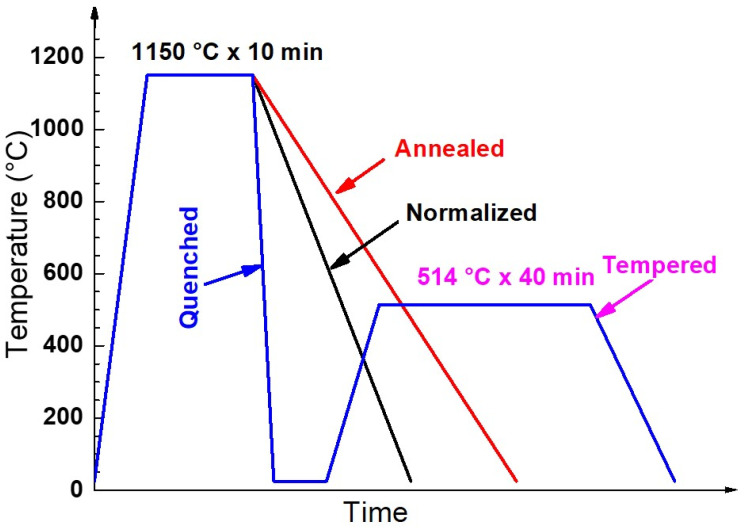
Schematic representation of the heat treatment schedules of 690 MPa grade steel used in the present work.

**Figure 2 materials-14-00851-f002:**
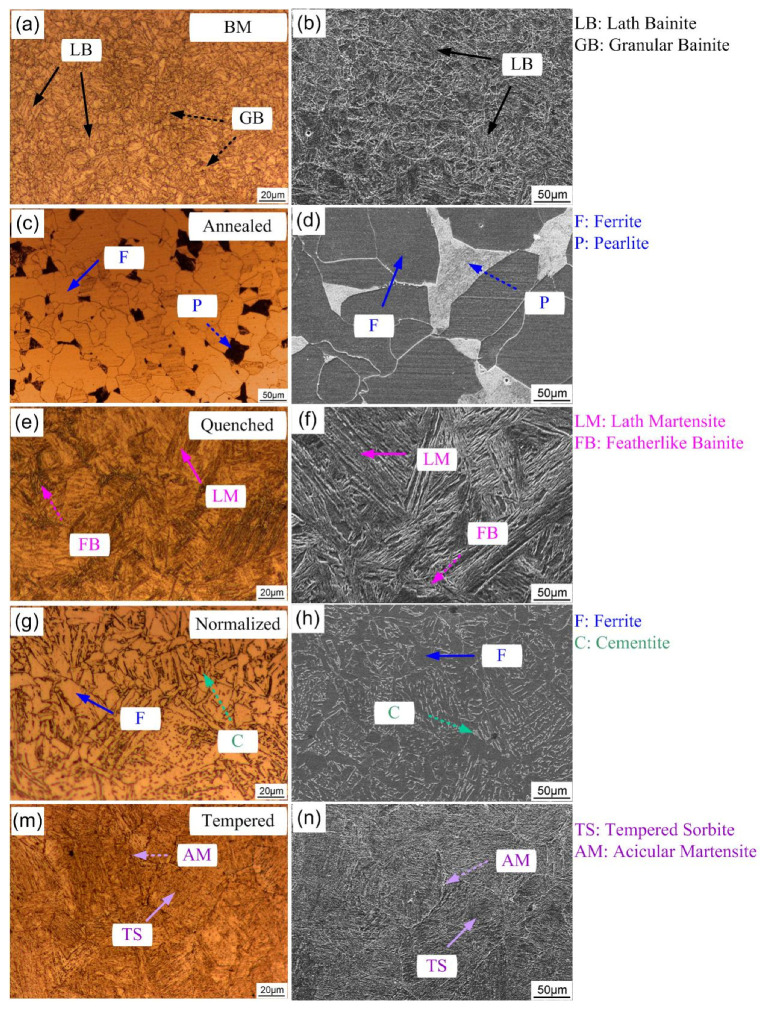
Microstructure characteristics of steel base metal (**a**,**b**), annealed steel (**c**,**d**), quenched steel (**e**,**f**), normalized steel (**g**,**h**), and tempered steel (**m**,**n**) observed by light microscope (**a**,**c**,**e**,**g**,**m**) and scanning electron microscope (SEM) (**b**,**d**,**f**,**h**,**n**).

**Figure 3 materials-14-00851-f003:**
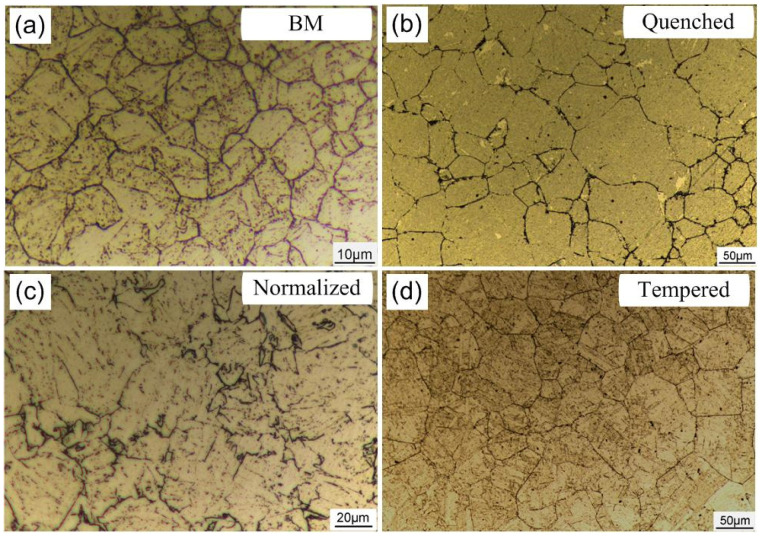
Metallographic diagram of the prior austenite grain boundaries of the base metal (BM) (**a**), quenched steel (**b**), normalized steel (**c**), and tempered steel (**d**).

**Figure 4 materials-14-00851-f004:**
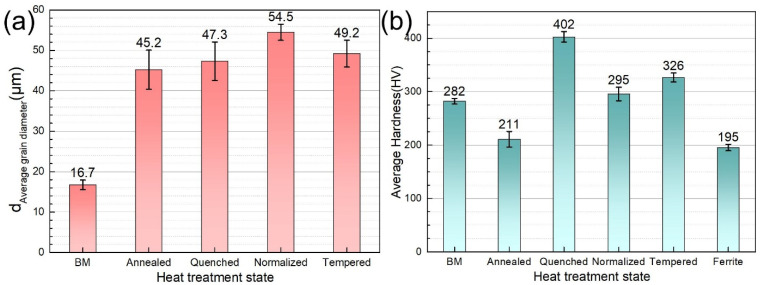
Average grain diameter (**a**) and Vickers microhardness (**b**) of 690 MPa grade marine engineering steel after different heat treatments and ferrite microstructure in the annealed steel.

**Figure 5 materials-14-00851-f005:**
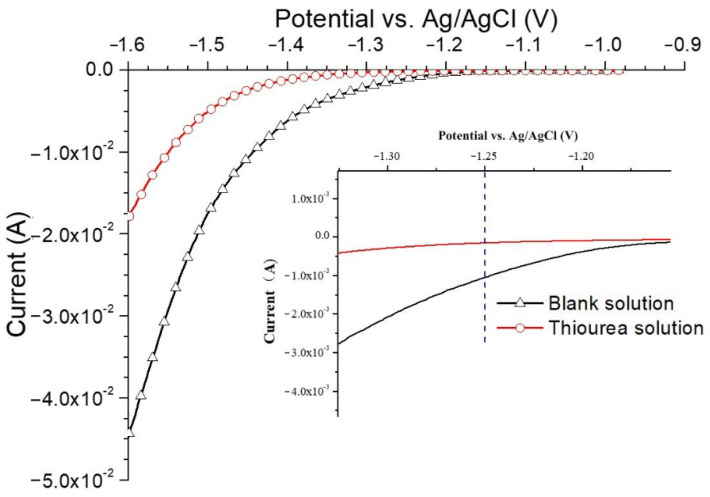
Linear sweep voltammogram between −0.9 V (vs. Ag/AgCl) and −1.6 V (vs. Ag/AgCl) for NaOH solution with and without thiourea and a larger version of the linear scanning volt-ampoule method (LSV) at −1.25 V (vs. Ag/AgCl) (scan rate 2 mV/s).

**Figure 6 materials-14-00851-f006:**
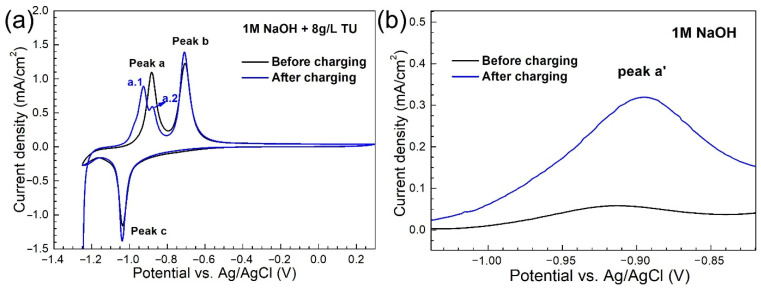
Cyclic voltammograms of the 690 MPa grade steel base metal obtained before and after hydrogen charging in NaOH solution with the thiourea (**a**) and without thiourea (**b**) (scan rate 10 mV/s).

**Figure 7 materials-14-00851-f007:**
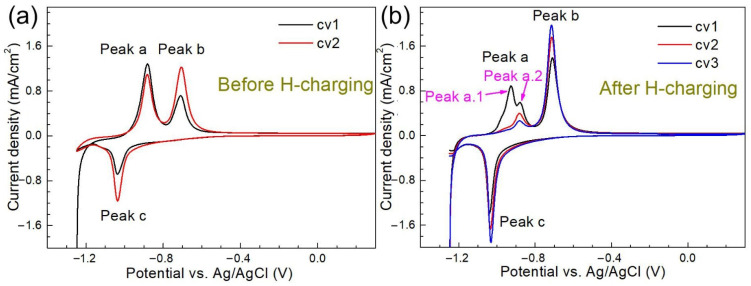
Cyclic voltammograms of the 690 MPa grade steel substrate before (**a**) and after H charging for 30 min (**b**) in NaOH solution containing thiourea (scan rate 10 mV/s).

**Figure 8 materials-14-00851-f008:**
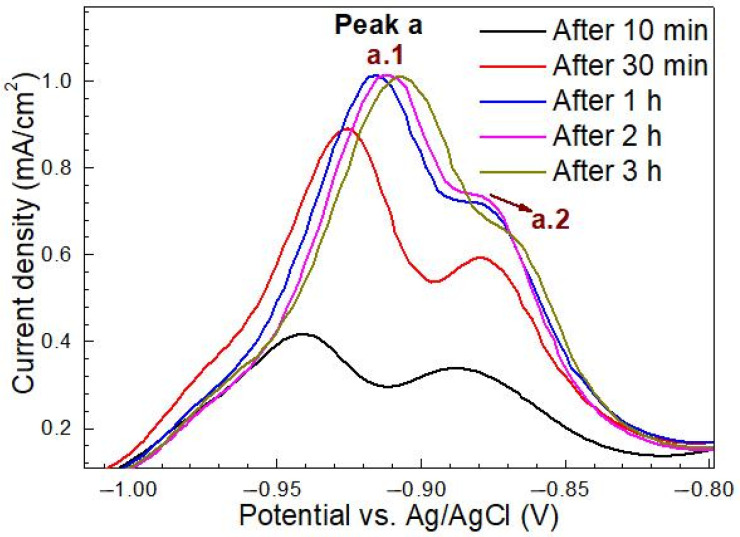
The first cyclic voltammetry (CV) scans for the base metal after hydrogen charging for 10 min, 30 min, 1 h, 2 h, and 3 h in NaOH solution containing thiourea (scan rate 10 mV/s).

**Figure 9 materials-14-00851-f009:**
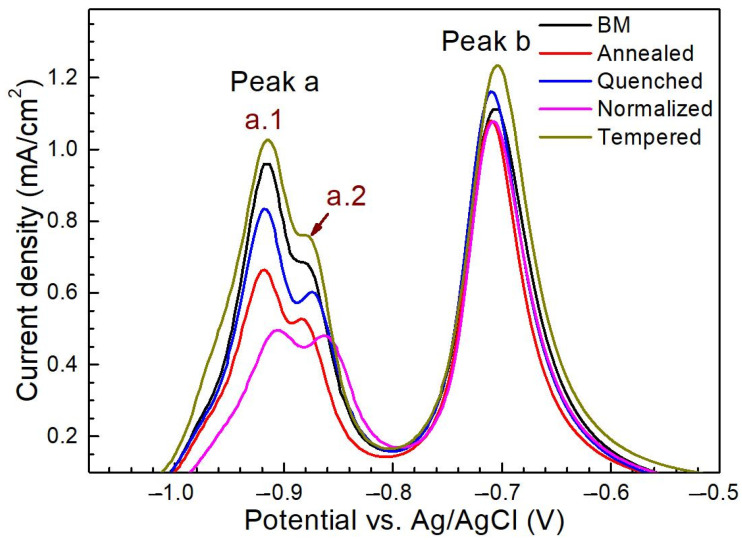
The first CV scans for the steels with different states after hydrogen charging for 30 min in NaOH solution containing thiourea.

**Figure 10 materials-14-00851-f010:**
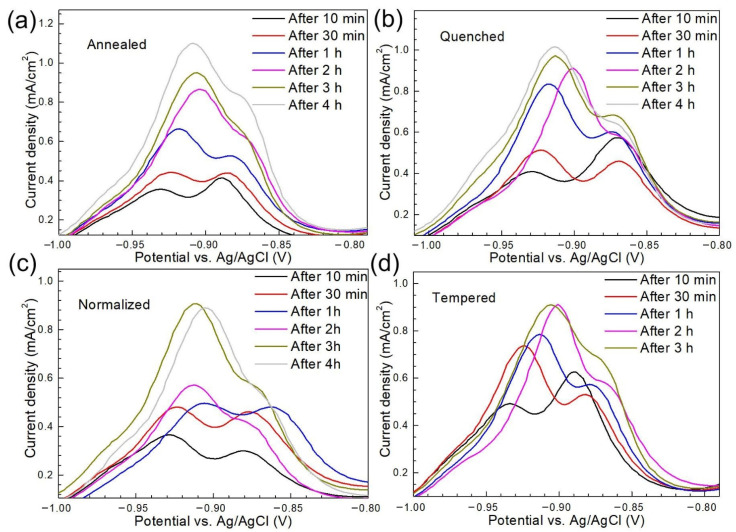
The first CV scans after hydrogen charging for different times in NaOH solution containing thiourea for annealed steel (**a**), quenched steel (**b**), normalized steel (**c**), tempered steel (**d**).

**Figure 11 materials-14-00851-f011:**
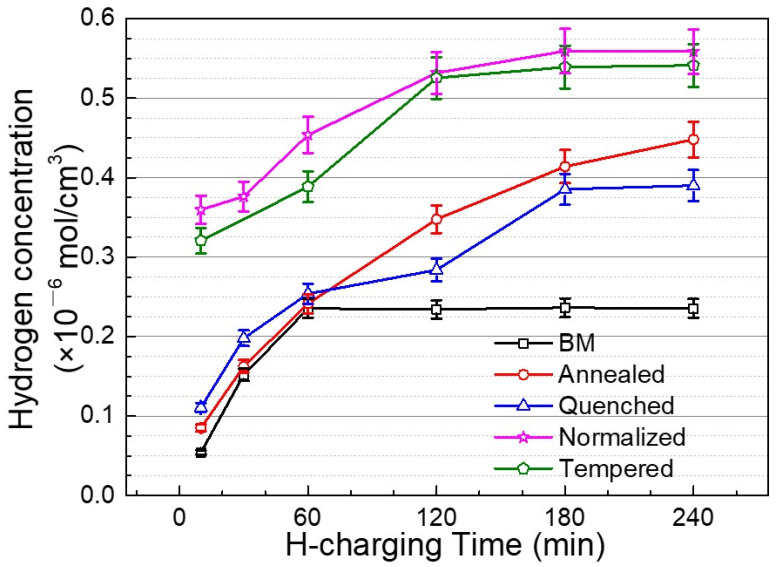
The absorbed H concentration released during potentiostatic hydrogen discharging (−0.9 V (vs. Ag/AgCl)) as a function of hydrogen charging time for the steel with different microstructures.

**Figure 12 materials-14-00851-f012:**
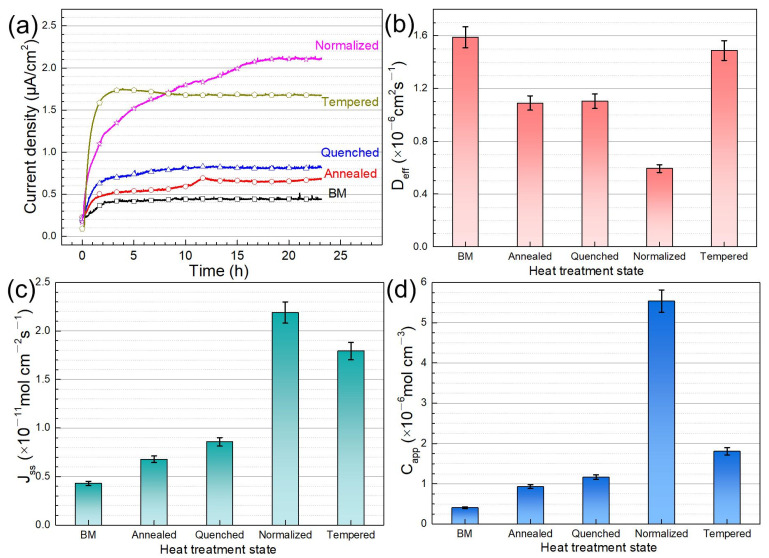
Hydrogen permeation curves (**a**) and the calculated parameters including *D_eff_* (**b**), *J_ss_* (**c**), and *C_app_* (**d**) of the steel with different microstructures in NaOH solution containing thiourea at −1.25 V (vs. Ag/AgCl).

**Figure 13 materials-14-00851-f013:**
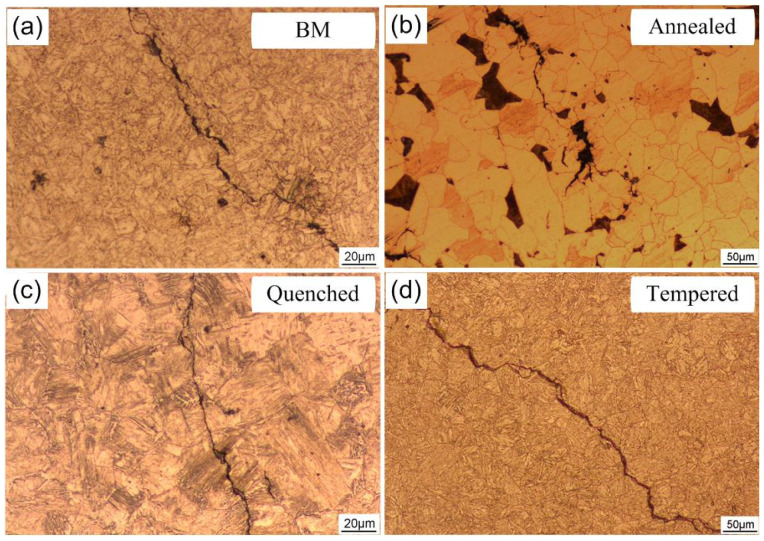
Light microscope micrographs of the hydrogen-induced cracks formed on the steel base metal (**a**), annealed steel (**b**), quenched steel (**c**), tempered steel (**d**).

**Figure 14 materials-14-00851-f014:**
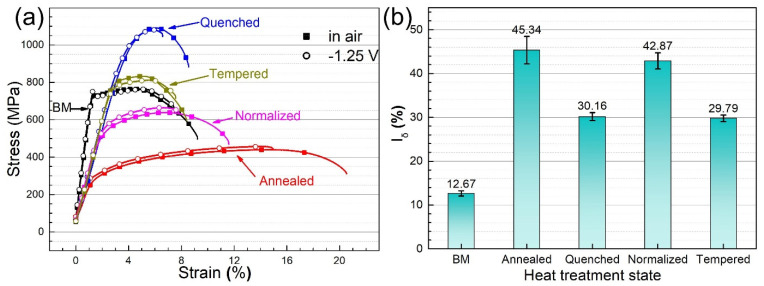
Stress–strain curves of the steel with different microstructures (**a**) and the corresponding hydrogen-induced cracking (HIC) susceptibility (**b**) in terms of the elongation loss rate in NaOH solution containing thiourea at −1.25 V (vs. Ag/AgCl).

**Table 1 materials-14-00851-t001:** Microstructure characteristics, test data, and cracking characteristics for BM and the heat treatment microstructure.

Material	Microstructure Morphology	Average Grain Diameter (μm)	The Absorbed H-Concentration (×10^−6^ mol cm^−3^)	*J* (mol cm^2^·s^−1^)	*C*_app_ (mol cm^−1^)	Crack Patterns	HIC Sensitivity
BM	Lath Bainite	15.9 ± 0.8	0.235 ± 0.012	0.429 ± 0.021	0.405 ± 0.020	Intergranular cracking	12.67 ± 0.61
Annealed steel	Ferrite and Pearlite	31.8 ± 1.6	0.448 ± 0.022	0.679 ± 0.034	0.935 ± 0.046	Distribution along the pearlite	45.34 ± 3.13
Quenched steel	Lath Martensite	63.5 ± 3.2	0.39 ± 0.020	0.859 ± 0.043	1.168 ± 0.058	Transgranular cracking	30.16 ± 0.91
Normalized steel	Low-Carbon Bainite	44.9 ± 2.2	0.559 ± 0.028	2.19 ± 0.110	5.537 ± 0.280	Transgranular cracking	42.87 ± 1.86
Tempered steel	Tempered Sorbite and Acicular Martensite	63.5 ± 3.2	0.541 ± 0.027	1.794 ± 0.090	1.808 ± 0.091	Intergranular cracking	29.79 ± 0.75

## Data Availability

Not applicable.

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
