# Peer review of "Correlation between Microstructure and Hydrogen Degradation of 690 MPa Grade Marine Engineering Steel"

_materials, 2021, doi:10.3390/ma14040851_

Round 1
Reviewer 1 Report
The paper aims at assessing the influence of hydrogen on the properties of 690 MPa grade marine steels characterized by different microstructures obtained with different heat treatments.
The paper is well written and clear and in the field of the “Materials” journal.
I would only suggest the following points:
General comment
The Authors should highlight and stress more the results obtained in the paper, since in some cases it not clear which results are obtained by the Authors and which are taken from the literature. For example, a Table summarizing the results obtained in the paper and the influence of the hydrogen on the investigated microstructures would increase the paper readability.
Abstract
- Page 1, line 18-19: “There are many internal defects in annealed steel and normalized steel, resulting in the formation of defects such as the two-phase interface which provides accommodation sites for H atoms and increases the HIC susceptibility”. This sentence is not clear and should be rewritten.
Introduction
- Page 1, line 29: “If the Ea of the trap..”. Ea is not defined in the text. All the symbols and the abbreviations should be defined in the text when they are used for the first time. Please check throughout the text.
Section 2
- Section 2.1: how were the temperatures for the heat treatments chosen? Were standard heat treatments considered in the present work? Please comment and clarify it.
Conclusions
- In Conclusions 1, the Authors state: “The hydrogen diffusion rate in the steel base metal with uniform microstructure and fine grain is the highest, while that in the annealed steel with ferrite and pearlite microstructure is the lowest”. In Conclusion 4, the Authors state: “HIC sensitivity of the base metal is the lowest due to its fine homogeneous grain structure, small hydrogen diffusion coefficient, and small hydrogen diffusion rate.” This seems to be contradictory: please revise the Conclusions and clarify it.
Minor comments
- Page 2, line 68: replace “are still studied” with “are in progress”.
- Page 4, line 166: replace “studied” with “was studied”.
Reviewer 2 Report
The manuscript presents original and technologically valuable results on 690 MPa grade steel substrate behavior in the presence of hydrogen. It is a highly important subject, keeping in mind growing interest in hydrogen-related technologies worldwide. Despite all benefits, the manuscript could be improved before publishing:
- There are lots of publications related to the hydrogen degradation of different steels. The novelty of the presented manuscript remains insufficiently justified. Clear remarks about novelty and added value to scientific understanding and technological progress would increase the value of the presented work.
- Some sentences remain not clear enough. For example, lines 67-70: “At present, a lot of studies have been carried out on the HIC behavior of microstructure, but researches are still studied on how to quantitatively analyze the relationship between microstructure and hydrogen.”. What authors would like to say? The presented work is also related to the correlation between the steel's microstructure and hydrogen degradation. Would be useful to clarify this sentence.
- The conclusion needs additional clarification. If to take “hydrogen diffusion rate” case Conclusion (1) says “ The hydrogen diffusion rate in the steel base metal with uniform microstructure and fine grain is the highest…” and Conclusion (3): “HIC sensitivity of the base metal is the lowest due to….small hydrogen diffusion rate”. I would also suggest not to use expressions like “small”. What does it mean small? Compare to what?
- Lines 100-101: “… optical microscope (OM)…” is mentioned. Would be useful to name the producer of this equipment.
- English language should be carefully checked. For example line 500: “So base matal..” (metal?)
Reviewer 3 Report
The work is prepared quite carefully. It contain typos and language errors and should therefore be carefully checked. I have two main considerations for this work. First of all, tempering of the 0.014C steel doesn't make much sense from a technological point of view. The results would be less surprising if they were for 0.14C steel. The characterization of the material was done quite carelessly and raises several doubts described below. Second, work does not have a significant new aspect to the discipline. The presented results, and especially their brief discussion, goes to confirming several elementary relationships. The authors do not introduce significant content on the mechanics and physics of the discussed phenomena, but only describe the facts noticed in the material characterization process. The study has features of an industrial analysis, not a scientific research. Not knowing the journal's policy, I leave the decision to the editor.
Additional remarks:
1 - in the footer we deal with MDPI Int. J. Environ. Res. Public Health, however, I don't know if it's an author or journal website negligence.
26 - Please provide an explanation of the HIC acronym with first ussage in the main text.
49 - "high strength is usually attributed to the presence of martensite". I suggest changing the phrase "martensite", because tempered martensite, for example, is not used technically. High-strength nanobainite steels cannot be ignored either. Maybe "structures formed in diffusionless transformations"?
56 - "microstructure, alloy content, second phase precipitation, grain structure, and martensite type" - "microstructure" is a truism that applies to each of the other features. I suggest adding the influence of individual variables on the HIC.
82 - "thick plate" - does it mean that each edge of the sample has been cut?
86 - What was the size of the samples? This is important in terms of the heat treatment time and research samples size. Please also provide information on the atmosphere in which the treatment was carried out.
98 - Due to the scope of work, I suggest changing the trade name of the shampoo to simply "detergent"
100 - I propose to change "optical microscope" to "light microscope" - SEM is also optical-based microscope
105 - HV measurement with such a small load and 5 measurements is not suitable for the comparison of mechanical parameters, which is confirmed in line 263, where the hardness does not match the expectations
126 - The electrolysis reaction pattern is wrong
227 - Bainitic structures in 0.014C steel? Didn't you confuse it with a fine ferritic microstructure, comminuted by factory thermomechanical treatment? Or could it be 0.14C, which however justifies the martensitic transformation, microstructure and hardness?
230 - "SEMpicture" - missing space
230 - while writing the entire chapter on shades in "SEM image", it is worth mentioning at least in chapter 2.2 that you used the SE electron imaging.
233 - Shouldn't 71 and 19 add up to 100? Does 19% perlite in the annealed 0.014C steel require a comment?
254 - There is no point in discussing the size of austenite grains in a quenched, normalized (annealed?), tempered sample. This parameter depends only on the material, temperature and austenitization time and should be consistent for the same parameters. If it is not, the authors either did not take care of the treatment process, or they deal with heterogeneous material, or took into account too small a statistical sample.
263 - Would you like to comment on the hardness of the annealed sample than the normalized one? This is against logic - either the heat treatment or the homogeneity of the material or the hardness measurement approach failed (a very low load and 5 measurements may not cover all phases sufficiently).
272 - The part concerning interaction with hydrogen is well prepared, but I am suspicious of the preparation of samples for the tests for which the above comments were described. Do they result from careless material tests, or has the material been processed with a certain randomness? This would undermine the whole point of the article.
426 - "Annealed steel has more active sites than other microstructures" - this sentence contradicts the previous one. The material after annealing has a structure most similar to the equilibrium structure, therefore mostly free from defects.
486 - The paragraph below refers to the research results and literature data but is not debatable - it does not discuss the similarities or differences in the above. It is very chaotic and lacks a clear description and justification of the mechanisms involved.
493 - too many phrases "hydrogen traps", it is worth reworking the text a bit
497 - "The microstructure of the base metal and normalized steel is similar" - a change in grain size by nearly 3x slightly contradicts this statement. Maybe "microstructure morphology"?
512 - "thiourea is an effective hydrogen permeation accelerator" - it is not a great progress compared to the work [DOI: 10.5006 / 0010-9312-35.12.535]
520 - "HIC sensitivity of the base metal is the lowest due to its fine homogeneous grain structure, small hydrogen diffusion coefficient, and small hydrogen diffusion rate" this sentence does not justify the fact of the most favorable HIC - diffusion properties are related to it, not its cause. Could you give any microstructure/defects/physics/energy explanation?
523 - "HIC sensivity (...)". The entire paragraph is a very modest summary of nearly 20 pages of work. The authors repeat the facts from their observations, and even the abstract already carried more justification and content.
Round 2
Reviewer 3 Report
Dear authors,
Thank you for responding quickly to my comments. I agree with most of explanations 1-8, 10, 12-13, 17-24. However, I have a few responses to comments:
In answer 14, the authors do not take into account that they describe two components of the microstructure with the proportions 71% and 19%, which adds up to 90% instead of 100%. Is there a component of the structure that I did not take into account?
In answer 15, the authors do not answer the question asked. I did not ask for a justification for the selection of the treatment temperature based on grain growth (which was presented very well), and the fact that in the paper they compare the austenite grain size for normalizing, hardening and annealing treatment. It is obvious that for the same treatment temperature of 1150 degrees and 10 minutes, the austenite structure should be similar, regardless of how the sample is cooled later. Why isn't it?
In answers 9 and 16 the authors (and also in the above 15) do not answer the objection concerning the results that do not correspond to the theory, and remove the results that do not match the theory. Of course, the reader will not be able to see these inconsistencies now, but does this mean that the material has been processed correctly and reproducibly? By the way, the results of hardness measurements are still included in the paper (Fig. 4b), despite the authors' assurances. The worse is that they are not described in the methodology.
Conducting (well-planned and conducted) analyzes on material that has been heat-treated and characterized without due attention allows the reader to doubt whether the obtained results really correspond to the research process being carried out.
Repeating the hardness measurements with the correct loading would be enough to reassure the reader that the microstructures correspond to the mechanical parameters. Cutting out the wrong methodology of hardness measurements from the text does not make the measurements correct. On the contrary, it introduces analysis without specifying the methodology.
